# Misaligned feeding impairs memories

**Dawn H Loh[1,2]\*, Shekib A Jami[2,3], Richard E Flores[1], Danny Truong[1], Cristina A Ghiani[1,4], Thomas J O'Dell[2,5], Christopher S Colwell[1,2]\***

[1]Department of Psychiatry and Biobehavioral Sciences, David Geffen School of Medicine, University of California, Los Angeles, Los Angeles, United States; [2]UCLA Integrative Center for Learning and Memory, University of California, Los Angeles, Los Angeles, United States; [3]Molecular, Cellular, and Integrative Physiology PhD Program, University of California, Los Angeles, Los Angeles, United States; [4]Department of Pathology and Laboratory Medicine, David Geffen School of Medicine, University of California, Los Angeles, Los Angeles, United States; [5]Department of Physiology, David Geffen School of Medicine, University of California, Los Angeles, Los Angeles, United States

**Abstract** Robust sleep/wake rhythms are important for health and cognitive function. Unfortunately, many people are living in an environment where their circadian system is challenged by inappropriate meal- or work-times. Here we scheduled food access to the sleep time and examined the impact on learning and memory in mice. Under these conditions, we demonstrate that the molecular clock in the master pacemaker, the suprachiasmatic nucleus (SCN), is unaltered while the molecular clock in the hippocampus is synchronized by the timing of food availability. This chronic circadian misalignment causes reduced hippocampal long term potentiation and total CREB expression. Importantly this mis-timed feeding resulted in dramatic deficits in hippocampal-dependent learning and memory. Our findings suggest that the timing of meals have far-reaching effects on hippocampal physiology and learned behaviour.

\*For correspondence: hloh@ucla. edu (DHL); CColwell@mednet. ucla.edu (CSC)

**Competing interests:** The authors declare that no competing interests exist.

## Introduction

The circadian system is a finely tuned network of central and peripheral oscillators headed by a master pacemaker, the suprachiasmatic nucleus (SCN), which governs daily rhythms in physiology and behaviour, including cognition. This network regulates cognitive processes (*Holloway and Wansley, 1973*; *Chaudhury and Colwell, 2002*), and the neural circuits involved in learning and memory also exhibit circadian rhythms in gene expression and synaptic plasticity (*Eckel-Mahan et al., 2008*; *Fropf et al., 2014*; *Lamont et al., 2005*; *Lyons, 2006*). Genetic disruption of these molecular oscillations has severe consequences on cognition (*Van der Zee et al., 2008*; *Wang et al., 2009*; *Wardlaw et al., 2014*). Environmental perturbations also have the capacity to disrupt synchrony and misalign this clock network (*Fekete et al., 1985*; *Cho et al., 2000*; *Devan et al., 2001*; *Ruby et al., 2008*; *Loh et al., 2010*; *Gibson et al., 2010*; *Karatsoreos et al., 2011*; *Fonken et al., 2012*; *LeGates et al., 2012*; *Fernandez et al., 2014*) and are problematic as many people in our modern society extend their work and recreation into the night hours.

There has been mounting evidence that the timing of when we eat is critical for our metabolic health (*Bass, 2012*; *Mattson et al., 2014*). At this point, timing of food intake is well-established to have a major impact on the phase of the molecular oscillations in peripheral organs such as the liver and pancreas (*Damiola, 2000*; *Stokkan, 2001*). Mis-timed meals during the sleep phase accelerates weight gain compared with animals fed during their wake phase (*Arble et al., 2009*; *Bray et al., 2013*), whereas wake-phase feeding has a protective effect against the cardiac and metabolic dysfunction caused by high fat diets (*Hatori et al., 2012*; *Gill et al., 2015*). Similar disruptive effects are

**eLife digest** Many processes within the body follow an approximately 24-hour cycle. In addition to patterns of sleep and wakefulness, such circadian rhythms help to regulate body temperature, blood pressure and hormone levels. They also affect when we feel hungry, when our muscles work most efficiently, and when we are mentally at our sharpest.

A region of the brain called the suprachiasmatic nucleus (SCN) generates and maintains circadian rhythms, and thus acts as the body's master clock. Daily exposure to light keeps the SCN synchronized with the 24-hour day/night cycle. However, most organs, from the heart to the pancreas, also possess their own clocks, which help to regulate organ-specific processes. These secondary clocks normally operate in synchrony with the SCN.

Exposure to light has long been known to influence circadian rhythms. However, more recent evidence suggests that the timing of meals may also affect circadian clocks, particularly those within the digestive system. Loh et al. therefore decided to investigate whether eating outside normal waking hours would also affect other key physiological processes, specifically the cognitive processes of learning and memory.

Mice normally consume most of their food after sunset. Loh et al. showed that rodents that were instead fed during the day performed less well on cognitive tests than other mice who received the same food at night. The daytime-fed mice showed changes in a region of the brain called the hippocampus, which supports learning and memory. In particular, daytime feeding changed the timing of the secondary circadian clock within the hippocampus, although it had no effect on the master clock in the SCN. Loh et al. therefore suggest that the misalignment of these circadian clocks impairs cognition.

Further experiments are needed to determine whether a similar relationship exists between the timing of meals and cognitive performance in humans. If so, these findings will have implications for the many individuals whose mealtimes, for work or social reasons, are out of synchrony with their body clocks.

seen in humans, where misaligned mealtimes produce cardiac and metabolic deficits, leading to a pre-diabetic state (*Scheer et al., 2009*). We thus became interested in the possibility that these ill consequences of eating at inappropriate phases of the daily cycle may also be maladaptive for cognitive function. In this study, we sought to determine the effects of chronic but stable misalignment of the circadian network by scheduling access to food at an inappropriate phase of the daily cycle. We demonstrate that this simple manipulation has far-reaching consequences for learning and memory.

## Results

### Misaligned feeding alters diurnal rhythms of activity and sleep

Mice were allotted a 6 hr window in which food was made available either during the middle of their active (aligned) or inactive (misaligned) phase (*Figure 1*). Mice adapted to the feeding protocol within 6 days (p = 0.9, *Figure 2A*) and there were no significant differences in body weight between the two groups at the time of testing (p = 0.5, *Figure 2B*). Daytime activity was increased in mice subjected to misaligned feeding (p < 0.001; *Figure 3A,B*), and the strength of daily rhythms of activity was reduced by misaligned feeding (p = 0.003; *Figure 3C*). Similarly, the temporal pattern of sleep was altered by misaligned feeding (*Figure 3D*). Immobility-defined video monitoring of sleep behaviour of misaligned mice showed decreased time spent asleep during the day (p < 0.001) and a corresponding increase in sleep during the night (p < 0.001; *Figure 3E*). Misaligned mice no longer exhibit a day vs night difference in sleep, sleeping equally during the day and night (p = 0.5). Crucially, the total time spent asleep over a 24 hr day was not reduced by misaligned feeding (p = 0.2). The change in temporal pattern of sleep quantity is also reflected in sleep fragmentation, where misaligned mice exhibit a greater number of sleep bouts (p < 0.05) with a corresponding decrease in average sleep bout duration (p < 0.001) during the day compared to aligned mice (*Figure 3—figure*

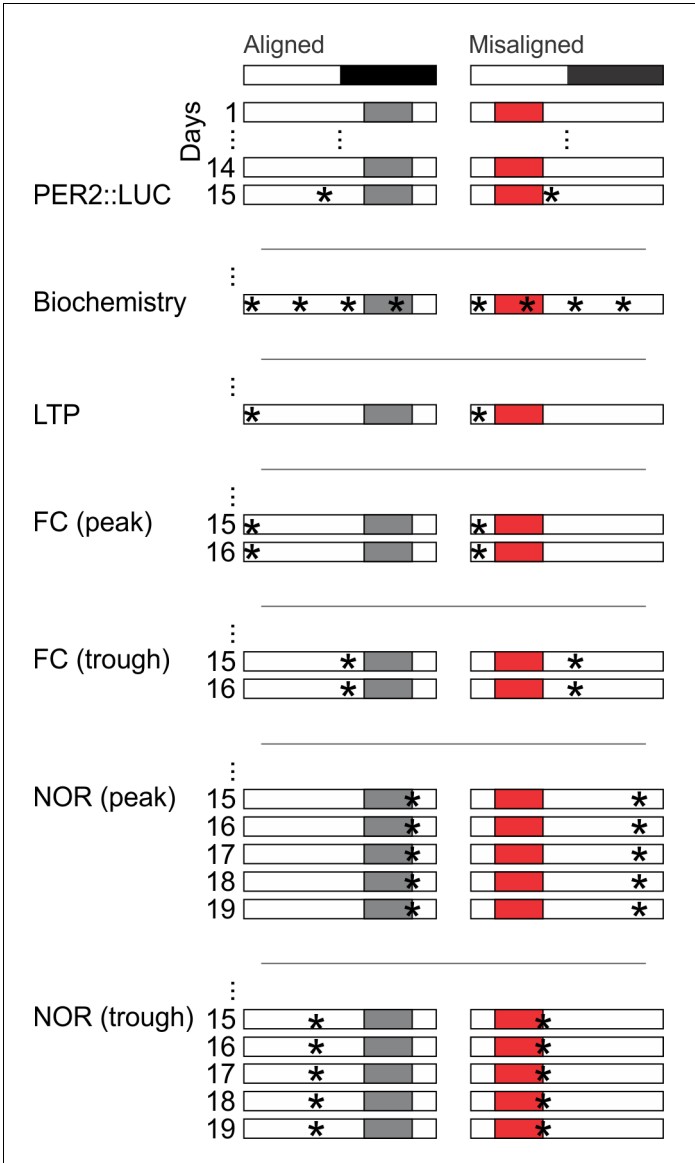

**Figure 1.** Cartoon schematic of experimental design. Standard mouse cages were modified to restrict access to the food chamber. Access was controlled by a motorized gate controlled by timed relay switches. Mice could only access food pellets when the gate was lifted (6 hr), and positive drive of the motor kept the gate closed for the remaining 18 hr. The scheduled feeding protocol was maintained for a minimum of 2 weeks prior to and during sample collections and behavioural tests, indicated by *. PER2::LUC, PER2-driven bioluminescence. LTP, long term potentiation. FC, fear conditioning. NOR, novel object recognition.

*supplement 1*). This increased day-time sleep fragmentation is compensated by fewer (p < 0.05) and longer (p < 0.001) night sleep bouts in the misaligned animals, resulting in no significant change in the total number (p = 0.9) and average duration (p = 0.6) of sleep bouts over a 24 hr period.

## Misaligned feeding alters phase of hippocampus without shifting the SCN

Using ex vivo organotypic cultures of explants from PER2::LUC reporter mice subjected to aligned or misaligned feeding, we confirmed that daytime feeding shifts the phase of the liver oscillator (p < 0.001) without altering the phase of the SCN oscillator (p = 0.07; *Figure 4A,C,D*). Importantly, we demonstrate that daytime feeding significantly misaligns the hippocampal oscillator by 14.1 hr (p <

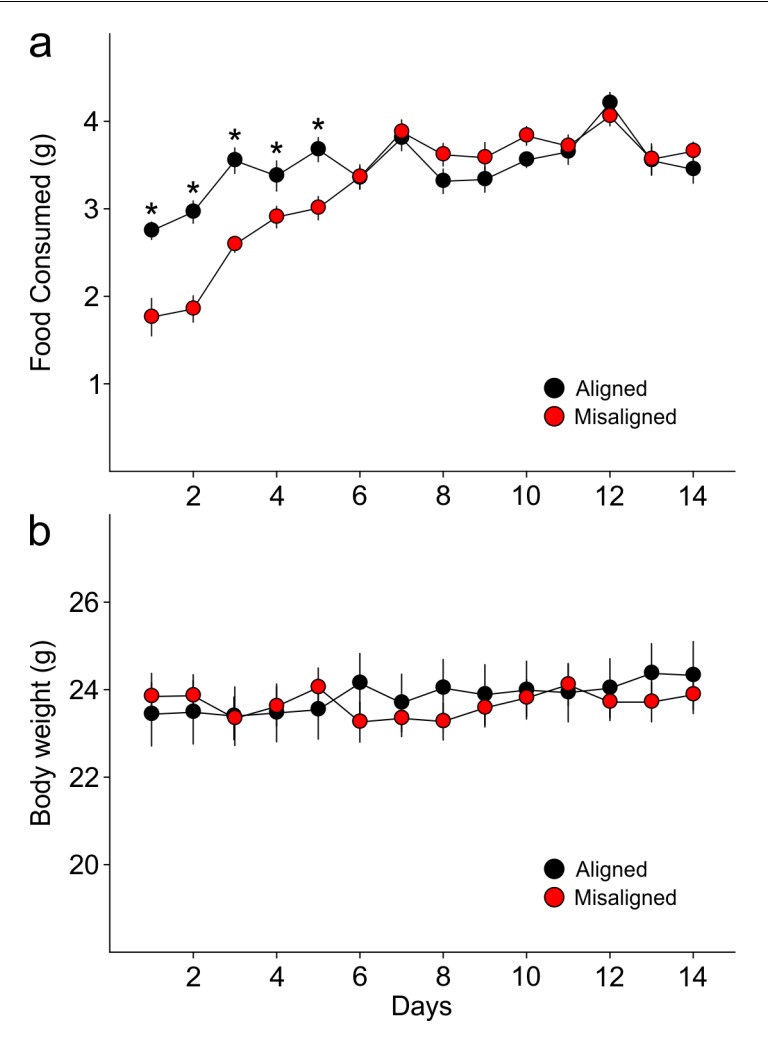

**Figure 2.** Food pellets in the automated feeding chambers were weighed daily to determine the amount of food consumed. (**a**) Misaligned mice caught up with aligned mice in daily food consumption by day 6 of scheduled feeding (p = 0.9) and did not differ in food consumption subsequently (day 6–14 *post hoc* p > 0.2). (**b**) Mice were weighed daily prior to food access. Body weights between treatment groups did not differ significantly through the duration of scheduled feeding (two way ANOVA p = 0.3). Line graphs represent the mean ± SEM (*n* = 16 per treatment).

The following source data is available for figure 2:

**Source data 1.** Food consumption and body weights of mice subjected to aligned and misaligned feeding.

---

0.001; *Figure 4B,D*), with small but significant effects on the intrinsic properties of the oscillator, including period (p = 0.05; *Figure 4E*) and damping rate (p = 0.02; *Figure 4F*).

## Misaligned feeding blunts total CREB expression and reduces hippocampal long term potentiation

We measured expression of phosphorylated CREB (pCREB) and total CREB (tCREB) in the hippocampus of aligned and misaligned mice at 6 hr intervals through a 24 hr day. pCREB protein levels were reduced in the misaligned animals at each time point, although significant differences were not detected (*Figure 5A,B*). Strikingly, expression of tCREB was significantly reduced by misaligned feeding (p < 0.01; *Figure 5A,C,D*), with the strongest effects in the day (*post hoc* p < 0.05 for ZT 2,

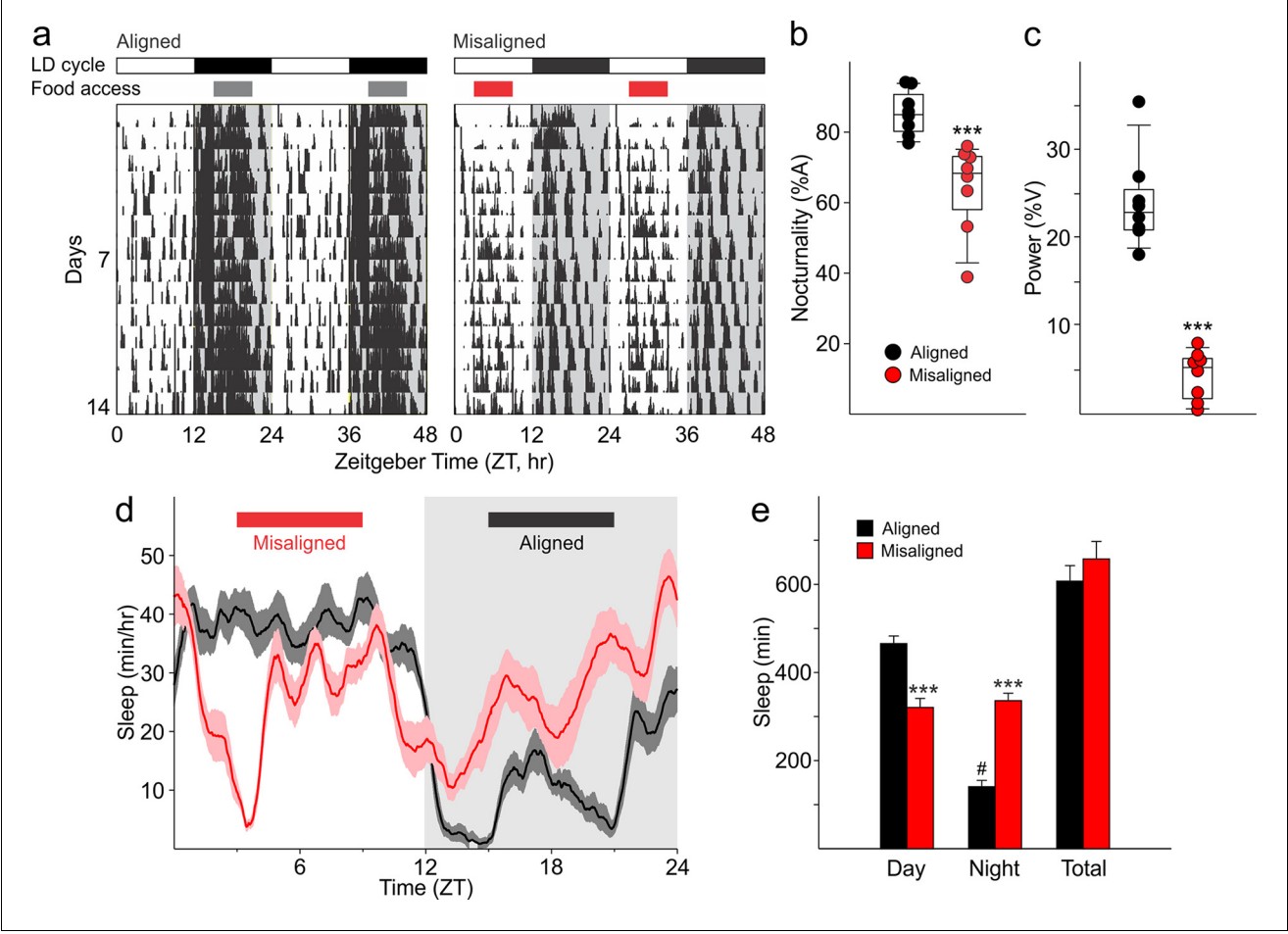

**Figure 3.** Altered temporal patterns of activity and sleep in mice subjected to misaligned feeding. (a) Mice (*n* = 8 per treatment) were given 6 hr windows of food access during the middle of the night (aligned, grey) or day (misaligned, red). Representative double-plotted actograms show the increased daytime activity of the misaligned mice throughout the treatment in a 12 hr:12 hr light:dark (LD) cycle. Grey shading in the actograms indicates lights off. (b) Nocturnality (% activity in the night) is reduced in misaligned mice (***p < 0.001). (c) Rhythm strength measured by the amplitude of a chi-square periodogram (%V) is reduced in misaligned mice (**p < 0.01). Box and whisker plots display the 25th to 75th percentiles, and the 10th to 90th percentiles respectively, with the median indicated by a line. (d) Sleep was measured by video monitoring after 2 weeks of scheduled food access (*n* = 10 per treatment). Running averages of immobility-defined sleep are shown for mice given aligned (black) and misaligned (red) access to food. The grey shading indicates lights off in a 12:12 LD cycle. (e) Total time spent asleep during the day (12 hr), night (12 hr), or over 24 hr are shown. ***p < 0.001 as determined by *t*-tests of aligned vs misaligned groups; # p < 0.001 day vs night within treatment. Bar graphs represent the mean ± SEM.

The following source data and figure supplement are available for figure 3:

**Source data 1.** Activity and sleep rhythm parameters of mice subjected to aligned and misaligned feeding.

**Figure supplement 1.** The temporal pattern of sleep fragmentation is altered by misaligned feeding without affecting overall sleep fragmentation over the 24 hr period.

8, and 20). This decrease in tCREB levels was uniformly observed throughout the hippocampus (*Figure 5D*).

Long term potentiation (LTP) responses of the dorsal hippocampus were measured during the day from mice subjected to aligned or misaligned feeding. LTP was significantly impaired in the misaligned group (p< 0.05; *Figure 5E*), indicating significant deficits in synaptic plasticity in the misaligned hippocampus. This decrease in LTP is specific to the misaligned feeding treatment and not due to an LTP-boosting effect by aligned feeding (*Figure 5—figure supplement 2*). To determine the impact of misaligned feeding on presynaptic neurotransmitter release probability, paired pulse

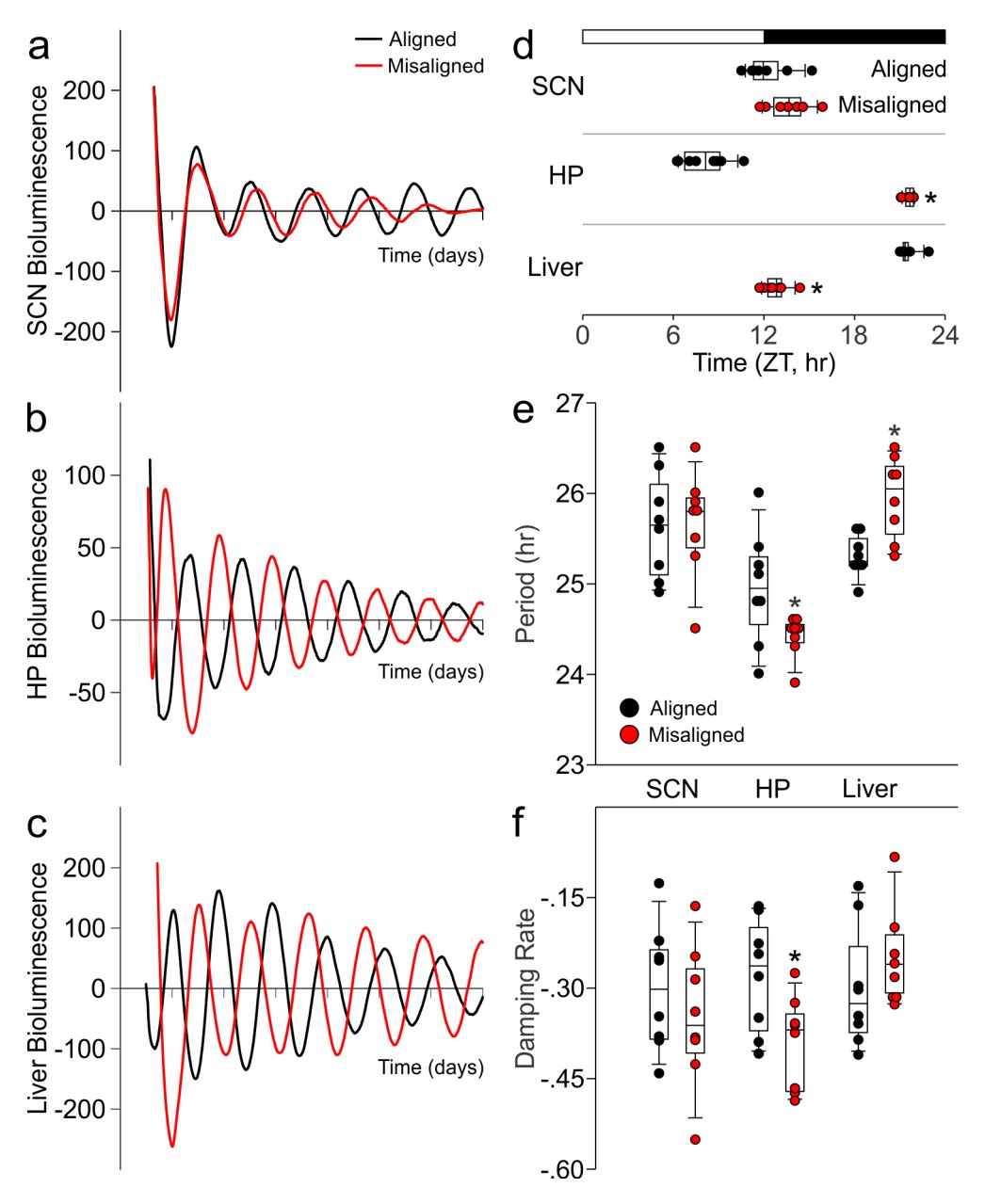

**Figure 4.** Differential impact of misaligned feeing on PER2-driven rhythms in bioluminescence of the SCN, hippocampus and liver (*n* = 8 per treatment). Representative examples of baseline-subtracted traces of PER2-driven bioluminescence in the SCN (**a**), hippocampus (HP; **b**), and liver (**c**) explants from aligned (black) and misaligned (red) mice. (**d**) Phase relationship between the first calculated peaks of ex vivo bioluminescence plotted against time of the prior lighting cycle (ZT) shows a significant phase change in the HP and liver. (**e**) Period of bioluminescence rhythms were determined by sine wave fitting. (**f**) Damping rates were determined from 6 days in culture. *denotes significant differences (p < 0.05) between aligned and misaligned samples. Box and whisker plots display the median as a line, the 25th to 75th percentiles, and the 10th to 90th percentiles respectively.

The following source data is available for figure 4:

**Source data 1.** PER2-driven bioluminescence rhythms of mice subjected to aligned and misaligned feeding.

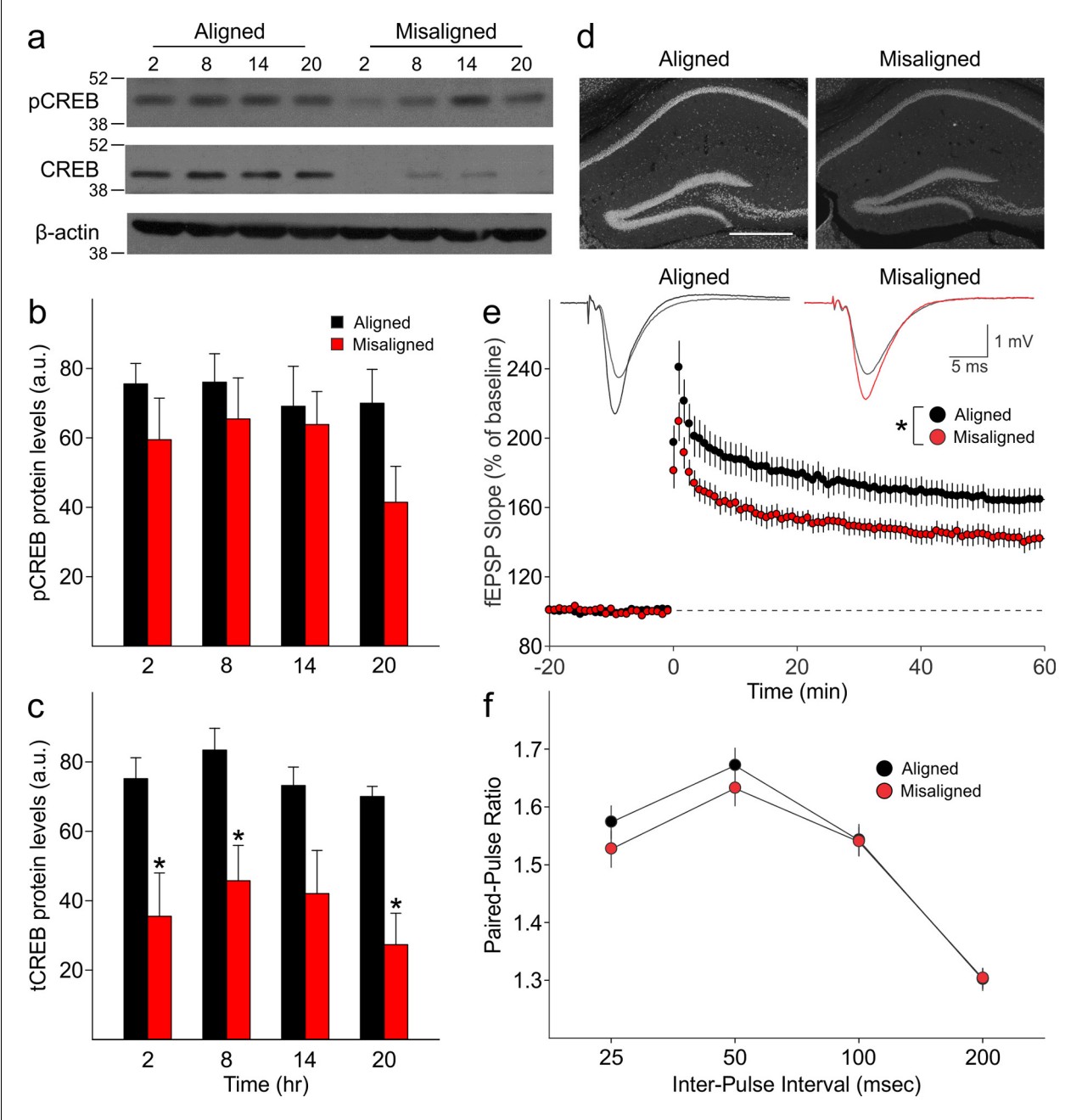

**Figure 5.** Reduced magnitude of hippocampal tCREB expression with a corresponding reduction in long term potentiation (LTP) in misaligned mice. (**a**) Representative immunoblots show the decrease in tCREB expression levels in misaligned vs. aligned mice, sampled at 6 hr intervals through the daily cycle (ZT). (**b**) pCREB levels are not significantly altered. (**c**) Misaligned feeding led to significant decreased levels of tCREB (p < 0.01). * indicates significant differences between groups at each time point (*post hoc* p < 0.05). Protein levels are expressed as arbitrary units (a.u.). Bar graphs represent the mean ± SEM of aligned (*n* = 4) and misaligned (*n* = 5) animals per time point. (**d**) tCREB immunoreactivity is decreased throughout the hippocampus in misaligned mice. Scale bar = 500 μm. (**e**) LTP was induced by high-frequency stimulation of the Schaffer collateral fibres, 2 x 100 Hz, 1 sec duration, 10 sec inter-train interval, delivered at time = 0. Daytime LTP responses recorded from the CA1 region were significantly decreased in misaligned mice (*n* = 6; p < 0.05) compared to aligned mice (*n* = 6). The inset shows fEPSPs recorded during baseline and 55–60 min post-HFS in aligned (left) and misaligned slice (right). (**f**) Paired-pulse facilitation ratios changed with intervals (p < 0.001), but were not significantly different between aligned and misaligned mice (p = 0.5) and no interactions between both factors were detected (p = 0.8). Line graphs represent the mean ± SEM (*n* = 6 per treatment).

The following source data and figure supplements are available for figure 5:

*Figure 5 continued on next page*

*Figure 5 continued*

**Source data 1.** Hippoca mpal CREB levels and LTP in mice subjected to aligned and misaligned feeding.

**Figure supplement 1.** Expression levels of *β*-actin did not vary with time of day in both groups (p = 0.9), nor between aligned and misaligned mice (p = 0.9).

**Figure supplement 2.** Aligned feeding does not significantly alter LTP magnitude compared to mice under *ad libitum* feeding (p = 0.4).

facilitation (PPF) experiments were performed at 25, 50, 100 and 250 msec intervals. There were no significant differences between the aligned and misaligned PPF ratios (p = 0.5; *Figure 5F*), and EPSP profiles were similar.

## Misaligned feeding affects hippocampal-dependent learning and memory

To test our hypothesis that misaligning the hippocampal oscillator from the SCN oscillator is detrimental to learning and memory, we subjected aligned and misaligned mice to hippocampal-dependent contextual fear conditioning. Mice were trained to associate a specific novel context to a fearful stimulus in the form of a mild shock. Both aligned and misaligned mice acquired freezing behaviour during the training trial (*Figure 6—figure supplement 1*). When tested 24 hr later by replacing the mice in the same context, the misaligned mice exhibited a significant reduction in fear-conditioned behaviour (ZT 2, p < 0.0001; *Figure 6A*, left), indicating that circadian misalignment affects long term memory. Performance on the fear conditioning test is dependent on time of day and performance peaks in the early day (*Chaudhury and Colwell, 2002*; *Eckel-Mahan et al., 2008*; *Loh et al., 2010*). To test for the possibility that misaligned mice have an inverted peak performance time, we trained and tested a separate cohort of mice at the opposite time of day (night, ZT 14). Both aligned and misaligned groups acquired freezing behaviour (*Figure 6—figure supplement 1*). Two way comparisons revealed a time of day effect on recall (p < 0.001) and an interaction between time of day and feeding condition (p < 0.001). Recall was significantly reduced in the aligned mice compared to the daytime tests (*post hoc* p < 0.001), but did not significantly change with time of day in misaligned mice (*post hoc* p = 0.3). Night-tested aligned and misaligned mice exhibited equally poor recall (*post hoc* p = 0.4; *Figure 6A*, right), ruling out the possibility that the misaligned mice have an altered peak phase of cognition.

We applied the novel object recognition test to examine cognition that peaks during the night in nocturnal rodents (*Ruby et al., 2008*). The misaligned mice exhibited significantly reduced novel object recognition (NOR) during night-time tests (ZT 21, p < 0.001; *Figure 6B*, right), indicating decreased cognitive performance. Day-time tests of a separate cohort of aligned and misaligned found no differences between both groups (*Figure 6B*, left), ruling out the possibility that peak performance of misaligned mice had shifted to the opposite phase. Two factor analysis revealed a significant effect of feeding condition (p = 0.003) and an interaction between time of day and feeding condition (p = 0.02). Specifically, aligned mice showed reduced NOR during the day (*post hoc* p = 0.04), and day-tested misaligned mice were not significantly different from aligned (*post hoc* p = 0.6).

## Discussion

Many people in our society find themselves working or playing during their normal sleep times. Due to these schedules, we eat around the clock with well-established literature of metabolic consequences (*Mattson et al., 2014*). In this study, we sought to determine if temporally restricted feeding schedules in mice could impact cognition. We found that time-restricted feeding led to dramatic impairments in learned behaviours such as hippocampal-dependent contextual fear conditioning and novel object recognition. Not all behavioural tests were similarly affected, as amygdala-dependent cued-fear conditioning was not altered by the misalignment. This implies that some learned behaviours are more vulnerable to the impact of misaligned feeding. Crucially, the total amount of sleep was not altered by the scheduled feeding and the mice did not lose weight as might be

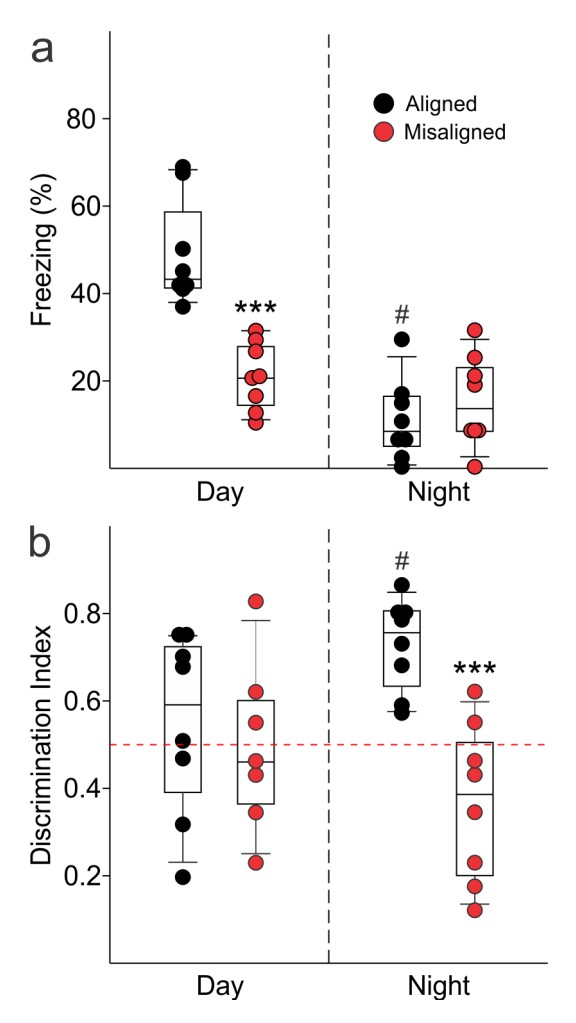

**Figure 6.** Memory deficits arise from misaligned feeding in mice. (**a**) Recall of the fear conditioned (FC) context is measured by the percentage of freezing when re-exposed to the fearful context. Misaligned mice ($n$ = 8) show significant deficits in recall of contextual (FC) compared to aligned mice ($n$ = 8;***p < 0.001) when trained and tested during the day (ZT 2). Circadian regulation of learning and memory is demonstrated by the decreased recall in aligned mice trained and tested at night (ZT 14; $n$ = 8; #p < 0.001). This time of day effect is lost in misaligned mice ($n$ = 8), which perform equally poorly at both times. (**b**) Novel object recognition (NOR) is reported using a discrimination index of $T_{novel}/(T_{novel}+T_{familiar})$, and mice are considered to exhibit NOR at values of 0.5 and above (dotted red line). NOR is impaired in misaligned mice trained and tested at night (ZT 21; $n$ = 8) compared to aligned mice ($n$ = 8; ***p < 0.001). Time of day effects were also found for NOR in aligned mice ($n$ = 8; #p < 0.01), which perform better during the night than day (ZT 9). Misaligned mice fail to show a time of day effect ($n$ = 7), again showing equally poor performance at both times. Box and whisker plots display the median as a line, the 25th to 75th percentiles, and the 10th to 90th percentiles respectively.

The following source data and figure supplement are available for figure 6:

**Source data 1.** Performance on long term memory tests in mice subjected to aligned and misaligned feeding.

**Figure supplement 1.** Acquisition of FC freezing in the day (ZT 2, $n$ = 8/group) and night (ZT 14; $n$ = 8/group) are unaltered by misaligned feeding.

expected if their caloric intake was restricted. Hence, by simply adjusting the time of food access, we demonstrated that we can alter the cognitive performance of mice. We do not know yet if this

equally applies to humans but shift work has been associated with decreased performance on cognitive tests (*Wright et al., 2006*; *Marquie et al., 2015*; *Zhou et al., 2011*).

The importance of sleep for memory is well-documented, where both the amount of sleep and the quality of sleep have been found to be critical for memory consolidation (reviewed in *Diekelmann and Born, 2010*). The misaligned feeding treatment did not result in an overall decrease in amount of sleep, but instead had a severe impact on the temporal pattern, suggesting that this treatment acts via disruption of the circadian timing of sleep. While sleep quality as assessed by polysomnography was not measured in this study, we were able to examine the degree of sleep fragmentation as determined by the number and duration of individual sleep bouts. Misaligned feeding again has a greater impact on the temporal pattern of sleep fragmentation, where the misaligned mice appear to "catch up" on consolidated sleep during what should be their active phase.

Mechanistically, we used the clock-driven rhythms in bioluminescence to demonstrate that the altered feeding time not only shifted the peripheral clocks in the liver, but also altered the circadian clock in the hippocampus. This finding is consistent with earlier literature suggesting the phasing of the molecular clock in the corticolimbic system and hippocampus are strongly influenced by the time of food availability (*Amir, 2004*; *Ángeles-Castellanos et al., 2007*; *Wakamatsu et al., 2001*). We have thus created a situation in which the molecular clocks within the nervous system are running at different phases, *i.e.* internal desynchronisation of the circadian system. A consequence of this misalignment is altered synaptic plasticity within the hippocampal circuit (Schaffer Collaterals/CA1) as revealed by the reduction in LTP. This is the first demonstration that the time of eating can impact the physiological underpinning of learned behaviour. Importantly, our manipulation of feeding caused a significant decline in CREB in the hippocampus, which has been demonstrated to be critical for memory allocation in mice (e.g. *Sano et al., 2014*; *Zhou et al., 2009*). The levels of tCREB were reduced at all phases that we sampled (ZT 2, 8, 14 and 20), and although we did not carry out cognitive tests at each of these phases, this finding indicates that memory would be impaired throughout the daily cycle. The reduction in tCREB provides a biochemical explanation for our finding, as well as providing a guidepost for future analysis of the effects of the misaligned feeding in the hippocampus. Therefore, this work raises the possibility that the timing of when we eat alters the physiological and biochemical events underlying learning and memory.

## Materials and methods

### Animals

All experimental protocols used in this study were approved by the UCLA Animal Research Committee (Protocol 1998–183). UCLA Division of Laboratory animal recommendations for animal use and welfare, as well as National Institutes of Health guidelines were followed. Adult (2–4 month old) male C57BL/6N wild-type mice (UCLA) were housed in a 12:12-hr lighting (LD) cycle with *ad libitum* access to food and water. For the bioluminescence experiments, PER2::LUC knock-in homozygotes on the C57Bl/6J background were used (*Yoo et al., 2004*). Mice were first entrained to a 12:12 LD cycle for a minimum of 2 weeks prior to further manipulations. For measurements of activity, sleep, and food consumption, mice were individually housed in cages. For bioluminescence measurements, learning and memory tests, hippocampal physiology, biochemistry and anatomy, mice were housed in groups of 3–5 mice per cage. Group-housed animals showed similar activity and food consumption patterns as singly housed animals. Both strains of mice exhibited similar activity and food consumption patterns.

### Scheduled food access

Mice entrained to a 12:12 LD cycle were randomly sorted into two groups per experiment. Cages were topped with wire grids adapted to restrict access to the food chamber, which enabled us to provide access to food during specific times of day (*Figure 1*). We visually verified that food pellets or powder were not being hoarded in nesting or bedding materials, but did not empty the cage bottoms to avoid excessive handling stress to the animals. Both groups were given a 6 hr window per 24 hr day to access the food chamber. Mice in one group were given access to the food chamber during the middle of the dark phase from Zeitgeber Time (ZT; ZT 0 equates to lights on) 15 to 21,

aligned to their active phase. Mice in the second group were given automated access to the food chamber during the middle of the light phase from ZT 3 to 9, misaligned from what should have been their active phase. Scheduled food access was applied for 2 weeks prior to and for the entire duration of all experiments. The exceptions were for the data described in *Figure 2* and *Figure 3A*: food consumption and activity, for which we monitored the mice from day 1 of scheduled food access.

## Activity monitoring

Mice were individually housed in automated feeding cages with a top-mounted passive infrared motion sensor (Honeywell IS-215T) to detect cage activity (aligned *n* = 8, misaligned *n* = 8). Data was collected using the Vital View system (Mini Mitter, Bend, OR) and analysed as previously described for wheel running activity using the El Temps software (*Loh et al., 2013*; A. Diez-Noguera, Barcelona, Spain). Specifically, nocturnality was calculated by determining the percentage of activity conducted during the dark phase (ZT 12–24) from 7 days of activity monitoring (days 8 to 14 of scheduled feeding). The strength of the daily rhythms was determined from the same 7 days, and is reported as the power of the circadian harmonic of a Fourier analysis (%V).

## Immobility-defined sleep measurements

Following activity monitoring, sleep-wake behaviour from the same cohort of aligned (*n* = 10) and misaligned (*n* = 10) mice was measured using continuous video recording and automated mouse tracking as previously described (*Loh et al., 2013*). Mice were maintained in the same automated feeding cages under the same lighting cycle and feeding schedule, and continuous video recording was performed from days 15 to 17 of scheduled feeding. Side-on views of the cage were acquired using CCTV cameras (Gadspot, GS-335C, City of Industry, CA), and the ANY-maze software (Stoelting Co., Wood Dale, IL) was used to track the animals. Prior work by Fisher and colleagues established the parameters for immobility detection (95% immobility of the area of the animal, for a minimum of 40 sec) by correlation with EEG/EMG recordings (*Fisher et al., 2012*). Immobility-defined sleep in 1 min bins from days 16 to 17 of scheduled feeding were averaged, and day (ZT 0–12) and night (ZT 12–24) sleep were compared. Average waveforms for display purposes were generated by smoothing the data using 1 hr running averages.

## PER2-driven bioluminescence activity in organotypic cultures

PER2::LUC male mice (2–3 mo) were subjected to either aligned (*n* = 8) or misaligned (*n* = 8) feeding for 2 weeks prior to sampling. Mice were sacrificed after anaesthesia (isoflurane) between ZT 10 and 11, and 1–2 $mm^3$ liver explants were immediately dissected in ice-cold Hanks' balanced salt solution (HBSS; Sigma, St Louis, MO) supplemented with 4.5 mM $NaHCO_3$, 10 mM HEPES and 100 U/ml penicillin-streptomycin as previously described (*Loh et al., 2011*). Brains were incubated in ice-cold slice solution (in mM: 26 $NaHCO_3$, 1.25 $NaH_2PO_4$, 10 glucose, 125 NaCl, 3 KCl, 5 $MgCl_2$, 1 $CaCl_2$) aerated with 95% $O_2$/5% $CO_2$ for 5 min, and 300 μm coronal sections were collected using a vibratome and further microdissected in HBSS under a 10X dissecting microscope. The SCN was cut away from the rest of the section using two cuts with a surgical scalpel (No. 21 blade, Fisher Sci., Waltham, MA). To acquire the hippocampal explant, the dorsal-most section (Bregma -1.2 to -1.6 mm) was microdissected to liberate the hippocampus and dentate gyrus by gently teasing away the cortex with scalpels (No. 11 blade, Fisher Sci.). All explants were individually transferred to Millicell membranes (0.4 μm, PICMORG50, Millipore, Bedford, MA) resting on 1.2 ml of recording media: (1X DMEM (Sigma), 1X B27 supplement (Gibco, Carlsbad, CA), 4.5 mM $NaHCO_3$, 10 mM HEPES, 40 mM Glutamax (Gibco), 4.5 mg/ml D-glucose, 25 U/ml penicillin, 25 U/ml streptomycin, 0.1 mM sodium salt monohydrate luciferin (Biosynth, Staad, Switzerland)) in a 35 mm dish sealed with autoclaved vacuum grease (Dow Corning, Midland, MI). SCN, hippocampus and liver explants were inserted into the Lumicycle photometer (Actimetrics, Wilmette, IL), incubated at 37°C, and bioluminescence was continuously monitored for 7 consecutive days. Raw bioluminescence values were normalized by baseline subtraction (24 hr running average) and smoothed with 2 hr windows to prepare the representative bioluminescence traces. The phase and damping rate of each explant were determined as previously described (*Loh et al., 2011*). Period was determined using the sine-wave fit function in Lumicycle Analysis (Actimetrics).

## Western immunoblotting

Hippocampi were rapidly dissected and homogenized in lysis buffer (50 mM Tris-HCl, 0.25% (w/v) sodium deoxycholate, 150 mM NaCl, 1 mM EDTA, 1% Nonidet P40, 1 mM sodium vanadate, 1 mM AEBSF, 10 ug/ml Aprotinin, 10 ug/ml Leupeptin, 10 ug/ml Pepstatin, and 1 mM sodium fluoride). Total protein concentration in cleared extracts was estimated using Pierce's BCA (bicinchoninic acid) Protein Assay Kit (Thermo Fisher Scientific, Carlsbad, CA) using bovine serum albumin as a standard. Western blots were performed as previously described (*Ghiani and Gallo, 2001*; *Ghiani et al., 2010*). Each extract was analysed for relative protein levels of phosphoCREB by using an antibody directed against the phosphorylated form at Ser133 (rabbit polyclonal, Millipore) and then for tCREB (rabbit polyclonal, Millipore). Equal protein loading was verified by Ponceau S solution (Sigma) reversible staining of the blots and each extract was also analysed for relative protein levels of β-actin (Sigma). Relative intensities of the protein bands were quantified by scanning densitometry using the NIH Image Software (Image J, http://rsb.info.nih.gov/ij/), and each value background-corrected. Data are shown as arbitrary units and are the average ± SEM of 4–5 animals/group. We confirmed by two way ANOVA comparisons that levels of β-actin do not vary with time of day in either group (p = 0.9) and is no different between aligned and misaligned mice (p = 0.9; *Figure 5—figure supplement 1*).

## Immunohistochemistry

Mice were perfused intracardially with 4% paraformaldehyde (PFA). Brains were dissected out, post-fixed in 4% PFA at 4°C overnight, cryoprotected in 15% sucrose. Immunolabelling of frozen sections (30 μm) was performed as previously described (*Ghiani et al., 2011*). Briefly, sections were blocked in carrier solution (1% BSA and 0.3% Triton X-100) containing 20% normal goat serum for 1 hr and incubated for 48h at 4°C with primary antibodies against tCREB (1:500 rabbit polyclonal, Millipore) diluted in carrier solution containing 5% normal goat serum. Sections were incubated with a goat anti-rabbit secondary antibodies conjugated to Cy3 (Jackson ImmunoResearch Laboratories, West Grove, PA) and mounted with Vectashield medium with DAPI (4′,6-diamidino-2-phenylindole; Vector Laboratories, Burlingame, CA). Immunostained sections were visualized using a Zeiss Axio Imager 2 with an AxioCam MRm. Images were acquired through Axiovision (Zeiss, Thornwood NY), using a 5X objective in order to visualize the entire hippocampus.

## Hippocampal slice preparation and electrophysiology

Hippocampal slice preparation and electrophysiological recordings were performed as previously described (*Carlisle et al., 2008*). Briefly, slices (400 μm) were maintained in oxygenated (95% $O_2$ / 5% $CO_2$), warmed (30°C) artificial cerebrospinal fluid (ACSF) containing 124 mM NaCl, 4.4 M KCl, 25 mM $NaHCO_3$, 1.0 mM $NaH_2PO_4$, 2.0 mM $CaCl_2$, 1.2 mM $MgSO_4$, and 10 mM glucose, and allowed to recover for at least 2 hr prior to the start of an experiment. A bipolar, nichrome wire stimulating electrode was placed in stratum radiatum of the hippocampal CA1 region to activate Schaffer collateral–commissural fibre synapses and an extracellular glass microelectrode filled with ACSF (resistance = 5–10 MΩ) was used to record evoked field excitatory postsynaptic potentials (fEPSPs). Extracellular recordings were done under interface conditions. The intensity of presynaptic fibre stimulation was adjusted to evoke fEPSPs with amplitude approximately 50% of the maximal fEPSP amplitude that could be elicited in each slice. fEPSPs were then elicited at 0.02 Hz throughout the experiment. LTP was induced by high-frequency stimulation (HFS, 2 trains of 100 Hz stimulation, 1 sec duration, inter-train interval = 10 sec). The average slope of fEPSPs normalized to baseline measured between 55 and 60 min post-HFS was used for statistical comparisons (two tailed *t* tests). The paired pulse stimulation pulses were delivered with interpulse intervals of 25, 50, 100 and 200 ms.

## Fear conditioning (FC)

Contextual fear conditioning was performed as previously described (*Chaudhury and Colwell, 2002*; *Wang et al., 2009*; *Loh et al., 2010*) with minor modifications. In brief, aligned and misaligned mice were tested and trained either in the day (ZT 2) or the night (ZT 14). Mice were trained and tested only once to avoid effects of prior manipulation. Results reported for daytime and nighttime tests are from distinct cohorts of animals. Mice were habituated to the testing room for 30 min under the relevant lighting conditions (dim red light, 2 lux, at night). The animals were individually

introduced to the novel environment (shock chamber) and allowed to familiarize to context for 3 min, after which the mice were trained to associate the context with a fearful unconditioned stimulus (US): foot-shock (0.2 mA). The training protocol consisted of 2 US with an inter-trial interval of 64 sec. At the end of the last US, mice were left in the chamber for a further 64 sec, after which they were returned to their home cages. 24 hr later, mice were placed individually into the same conditioning chamber for 6 min. Video recordings of the acquisition and recall tests were done with CCTV (Gadspot) cameras with supplemental infrared lighting during both times of day. Freezing behaviour was scored as previously described (*Chaudhury and Colwell, 2002*). Cued fear conditioning was performed as previously described (*Wang et al., 2009*). Using the same stimulus protocol as contextual FC, 30 sec of white noise (80 dB; conditioned stimulus, CS) preceded the presentation of US1 and US2. 24 hr later, mice were placed individually into a novel testing chamber. Following 2 min of baseline, the cued tone was presented for 2 min; mice were left in the chamber for a further 2 min. Freezing behaviour was scored as previously described.

## Novel object recognition (NOR)

Habituation to the testing arena, object familiarization, and testing for NOR over 5 consecutive days were performed in the active phase at ZT 21 ($n = 8$ per feeding condition) for night-time tests under dim red light (2 lux), with day tests at ZT 9 ($n = 8$ per feeding condition). Mice were habituated to the testing arena (60 $\times$ 48 $\times$ 30 cm) in 10 min trials on 2 consecutive days. During familiarization trials, two identical objects were placed equidistant from each other and the walls of the arena, and mice were allowed to explore the arena for 10 min on 2 consecutive days. Mice that failed to travel 20 m or interact with objects for more than 20 sec in the second familiarization trial were eliminated from further analysis. On the day of testing for NOR, one familiar object was replaced with a novel object with a different shape and made of a different material (plastic or glass). The testing trial was 5 min in duration. Arenas and objects were wiped between animals with 10% Windex and dried using paper towels. Video feeds of each arena from an overhead CCTV camera supplemented with infrared lighting were fed to the ANY-maze software. The arena and object maps were defined in ANY-maze, and allowed for automated tracking of the animal's area, with the ability to distinguish between the head and tail. Entry of the animal's head to the defined object's area was scored for time interacting with the object. An experimenter watched the recorded videos and overlaid tracking in real-time to verify that tracking of the animal's area and head/tail orientation was consistent and that scoring of object interaction was accurate. Performance on the NOR test is indicated by a discrimination index calculated of time spent with the novel object ($T_{novel}$) divided by the sum of time spent with both objects ($T_{novel}+T_{familiar}$). One animal from the misaligned group (out of 8) tested during the day failed to sufficiently explore both objects (< 10 sec during the second familiarization trial) and was eliminated from further analysis. For all novel object recognition experiments, mice were trained and tested only once to avoid effects of prior manipulation. Results reported for day-time and night-time tests are from distinct cohorts of animals.

## Statistical analysis

All statistical analysis was performed using Sigma Stat (ver. 12). Effects of misaligned feeding on activity, sleep, PER2-driven bioluminescence, and LTP were determined using unpaired *t*-tests in comparison to the aligned group. In cases of unequal variance, the Mann-Whitney rank sum test was applied. To determine the effects and interaction of time of day and scheduled feeding condition on protein expression, and hippocampal-dependent learning and memory, we applied two factor analysis of variance (ANOVA) tests. The Holm-Sidak *post hoc* test was used to distinguish differences due to misaligned feeding. Differences with p < 0.05 were deemed significant in all analyses. Where appropriate (*Figure 3,4,6*), box plots indicate the 25th and 75th percentiles, with error whiskers indicating the 10th and 90th percentiles. Values in the text, line graphs, and bar graphs are reported as mean ± standard error mean (SEM).

## Acknowledgements

We thank the following for their invaluable assistance: John Parker (Physiology, UCLA) for fabrication of automated feeding machines to schedule food access, Analyne Schroeder and Dika Kuljis for assistance with the bioluminescence organotypic cultures, Yingfei Wu for assistance with

immunohistochemistry, Collette Kokikian and Karen Cheng for assistance with cognition tests, and Donna Crandell (Intellectual Disorders and Disabilities Research Center, UCLA) for assistance with the graphics.

## Additional information

### Funding

| Funder | Grant reference number | Author |
|---|---|---|
| National Institute of General Medical Sciences | NRSA training grant 2T32GM065823 | Shekib A Jami |
| National Institute of General Medical Sciences | MARC training grant 2T34GM008563 | Richard E Flores |
| O'Keefe Foundation | | Christopher S Colwell |
| National Institute of Child Health and Human Development | P30HD004612 | Dawn H Loh Cristina A Ghiani Christopher S Colwell |
| National Institute of Child Health and Human Development | U54HD087101 | Dawn H Loh Cristina A Ghiani Christopher S Colwell |

The funders had no role in study design, data collection and interpretation, or the decision to submit the work for publication.

### Author contributions

DHL, CAG, TJO'D, Conception and design, Acquisition of data, Analysis and interpretation of data, Drafting or revising the article ; SAJ, Acquisition of data, Analysis and interpretation of data, Drafting or revising the article; REF, DT, Acquisition of data, Analysis and interpretation of data; CSC, Conception and design, Drafting or revising the article

### Author ORCIDs

Dawn H Loh, http://orcid.org/0000-0001-7876-5757
Cristina A Ghiani, http://orcid.org/0000-0002-9867-6185

### Ethics

Animal experimentation: This study was performed in strict accordance with the recommendations in the Guide for the Care and Use of Laboratory Animals of the National Institutes of Health. All of the animals were handled according to approved institutional animal care and use committee (IACUC) protocols of the University of California Los Angeles (Protocol 1998-183).

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
