## [Decision Letter]

Thank you for submitting your work entitled "Midnight snacking mangles memories" for peer review at *eLife*. Your submission has been favorably evaluated by a Senior editor and three reviewers, one of whom, Joseph S. Takahashi, is a member of our Board of Reviewing Editors. One of the three reviewers, Daniel Storm, has also agreed to share his identity.

The reviewers have discussed the reviews with one another and the Reviewing editor has drafted this decision to help you prepare a revised submission.

Summary:

The manuscript by Loh et al. presents a carefully conducted and important series of experiments uncovering the consequences of mistimed feeding on the hippocampal circadian clockwork and learning and memory. The main findings indicate that feeding during the inactive phase of the circadian cycle leads to ~14 h phase alteration in hippocampal clock gene rhythms whereas SCN phase is unaltered. These changes were associated with deficits in hippocampal LTP, total hippocampal CREB expression, and hippocampal-dependent learning and memory.

This is an excellent paper and is consistent with a number of other studies that have established that memory consolidation depends on the circadian rhythm. This is very interesting because it extends our understanding of circadian rhythm and memory by showing that the timing of heating affects hippocampus-dependent neuroplasticity. The experiments are well done and the major conclusions are justified by the data.

Essential revisions:

1) I understand why the authors chose a comparison of mice fed during the day to mice with food restricted to the dark phase. This not only controls for time of food presentation but also more accurately mimics the human condition. However, under laboratory conditions, mice eat throughout the day with food consumption peaking during their active phase. The possibility might be considered that restricting feeding to the active phase enhances LTP, learning and memory, etc. above that normally seen in mice either by surveying the literature or adding a group of ad lib fed animals as a point of comparison.

2) Because hippocampal rhythms are 14.1 h out of phase between night- and day-fed animals, testing animals on the fear and novel object tasks 12 hours apart might not be the ideal timing to control for the possibility that day-fed animals are being examined at an inappropriate time for learning potential. The CREB data suggest this is not the case, but the point might be considered in the manuscript.

3) Although the total amount of sleep measured by inactivity is the same across day- and night-fed mice, as the authors acknowledge, the pattern is quite different. The possibility should be considered that the pattern/quality of sleep might contribute to the deficits observed, particularly as EEG data are unavailable and it's unknown whether sleep quality was markedly impacted. Given the importance of sleep for memory consolidation, this point is of particular importance.

4) Although the title is creative, it is incongruent with the results as mice eating at midnight would be eating at their ideal time of day. The title should be revised to be informative and consistent with the contents of the paper.

5) In the subsection “Animals”, the specific C57BL/6 substrain is not specified, nor is the supplier. This is important because the two most common substrains, C57BL/6J and C57BL/6N, differ in behavioral tests, and the C57BL/6N strain carries a number of known mutations compared to the C57BL/6J strain (Kumar et al. 2013 Science 342:1508). In previous publications from the Colwell lab, the C57BL/6 mice were obtained from Charles River, which would make the correct mouse strain nomenclature, C57BL/6NCrl.

6) In the subsection “Activity monitoring”, “power" should not be estimated using a Chi-square periodogram because this method operates in the time domain, therefore, it cannot independently assess power at different frequencies. The appropriate method is to use a Fourier analysis method to assess power in the circadian range.

7) In the subsection “Fear conditioning (FC)”, for context-dependent fear conditioning experiments, the methods indicate that the mice were introduced to the novel environment for 15 minutes before the US. This is an unusually long time and is not conventional in this assay. In the cited references from the authors' labs (Chaudhury and Colwell, 2001; Loh et al., 2010), the methods in these papers specify a 3-minute time in the novel environment, which is the typical time used by the vast majority of studies, and has been shown long ago to be the optimal time interval.

---

## [Author Response]

*Essential revisions:*

*1) I understand why the authors chose a comparison of mice fed during the day to mice with food restricted to the dark phase. This not only controls for time of food presentation but also more accurately mimics the human condition. However, under laboratory conditions, mice eat throughout the day with food consumption peaking during their active phase. The possibility might be considered that restricting feeding to the active phase enhances LTP, learning and memory, etc. above that normally seen in mice either by surveying the literature or adding a group of ad lib fed animals as a point of comparison.*

We found no significant differences in LTP magnitude of C57Bl/6J mice under either ad libitum feeding or the aligned feeding treatment (*P* = 0.4 at 60 minutes post-stimulation). We show this data now in Figure 5—figure supplement 2.

*2) Because hippocampal rhythms are 14.1 h out of phase between night- and day-fed animals, testing animals on the fear and novel object tasks 12 hours apart might not be the ideal timing to control for the possibility that day-fed animals are being examined at an inappropriate time for learning potential. The CREB data suggest this is not the case, but the point might be considered in the manuscript.*

The levels of tCREB were reduced at all phases that we sampled. Although we did not carry out learning tests at all phases, the tCREB data indicate that memory would be impaired throughout the daily cycle. We now make this point in our Discussion as recommended:

“Importantly, our manipulation of feeding caused a significant decline in CREB in the hippocampus, which has been demonstrated to be critical for memory allocation in mice (e.g. Sano et al., 2014; Zhou et al., 2009). The levels of tCREB were reduced at all phases that we sampled (ZT 2, 8, 14, and 20), and although we did not carry out cognitive tests at each of these phases, this finding indicates that memory would be impaired throughout the daily cycle.”

*3) Although the total amount of sleep measured by inactivity is the same across day- and night-fed mice, as the authors acknowledge, the pattern is quite different. The possibility should be considered that the pattern/quality of sleep might contribute to the deficits observed, particularly as EEG data are unavailable and it's unknown whether sleep quality was markedly impacted. Given the importance of sleep for memory consolidation, this point is of particular importance.*

Although we did not perform EEG recordings, and are unable to definitively report on sleep quality, we were able to distinguish the number and duration of individual immobility-defined sleep bout. We have added a supplementary figure to Figure 3 to show that the number and duration of sleep bouts in the day, as a measure of sleep fragmentation, are affected by misaligned feeding. Intriguingly, this measurement of sleep fragmentation is not significantly different between aligned and misaligned mice when considered over the 24 hr day and night, and the number of sleep bouts of long durations (>30 minutes) is equivalent between the two treatment groups. This suggests that the misaligned mice may experience sleep compensation, and have shifted “consolidated” sleep to the nighttime. A paragraph has been added to the Discussion to address this:

“The importance of sleep for memory is well-documented, where both the amount of sleep and the quality of sleep have been found to be critical for memory consolidation. The misaligned feeding treatment did not result in an overall decrease in amount of sleep, but instead had a severe impact on the temporal pattern, suggesting that this treatment acts via disruption of the circadian timing of sleep. While sleep quality as assessed by polysomnography was not assessed in this study, we were able to examine the degree of sleep fragmentation as determined by the number and duration of individual sleep bouts. Misaligned feeding again has a greater impact on the temporal pattern of sleep fragmentation, where the misaligned mice appear to ‘catch up’ on consolidated sleep during what should be their active phase.”

*4) Although the title is creative, it is incongruent with the results as mice eating at midnight would be eating at their ideal time of day. The title should be revised to be informative and consistent with the contents of the paper.*

We have changed the title of the manuscript to more accurately reflect the relative time of food intake: “Misaligned feeding impairs memories”.

*5) In the subsection “Animals”, the specific C57BL/6 substrain is not specified, nor is the supplier. This is important because the two most common substrains, C57BL/6J and C57BL/6N, differ in behavioral tests, and the C57BL/6N strain carries a number of known mutations compared to the C57BL/6J strain (Kumar et al. 2013 Science 342:1508). In previous publications from the Colwell lab, the C57BL/6 mice were obtained from Charles River, which would make the correct mouse strain nomenclature, C57BL/6NCrl.*

The WT mice used in this study were obtained from a colony maintained at UCLA, which was originally derived from the C57Bl/6N strain. The PER2::LUC knock-in transgenic mice were backcrossed to the C5Bl/6J strain. Both strains of mice exhibited the same alterations in activity and sleep rhythms due to the scheduled food access.

*6) In the subsection “Activity monitoring”, "power" should not be estimated using a Chi-square periodogram because this method operates in the time domain, therefore, it cannot independently assess power at different frequencies. The appropriate method is to use a Fourier analysis method to assess power in the circadian range.*

We now report the power of the circadian harmonic of a Fourier analysis.

7) In the subsection “Fear conditioning (FC)”, for context-dependent fear conditioning experiments, the methods indicate that the mice were introduced to the novel environment for 15 minutes before the US. This is an unusually long time and is not conventional in this assay. In the cited references from the authors' labs (Chaudhury and Colwell, 2001; Loh et al., 2010), the methods in these papers specify a 3-minute time in the novel environment, which is the typical time used by the vast majority of studies, and has been shown long ago to be the optimal time interval.

This was a mistake in the text, which has been corrected. We used a 3-minute context familiarization followed by the 2 US protocol.